# Systematic Analysis of a Military Wearable Device Based on a Multi-Level Fusion Framework: Research Directions

**DOI:** 10.3390/s19122651

**Published:** 2019-06-12

**Authors:** Han Shi, Hai Zhao, Yang Liu, Wei Gao, Sheng-Chang Dou

**Affiliations:** 1School of Computer Science and Engineering, Northeastern University, Shenyang 110169, China; hai.zhao.310@gmail.com; 2School of Information Science and Engineering, Shenyang University of Technology, Shenyang 110870, China; wdly1219@163.com; 3College of Computer Science and Technology, Shenyang University of Chemical Technology, Shenyang 110142, China; gwe_mail@163.com

**Keywords:** Internet of Battlefield Things, Body Sensor Networks, wearable device, information fusion, sensor

## Abstract

With the development of the Internet of Battlefield Things (IoBT), soldiers have become key nodes of information collection and resource control on the battlefield. It has become a trend to develop wearable devices with diverse functions for the military. However, although densely deployed wearable sensors provide a platform for comprehensively monitoring the status of soldiers, wearable technology based on multi-source fusion lacks a generalized research system to highlight the advantages of heterogeneous sensor networks and information fusion. Therefore, this paper proposes a multi-level fusion framework (MLFF) based on Body Sensor Networks (BSNs) of soldiers, and describes a model of the deployment of heterogeneous sensor networks. The proposed framework covers multiple types of information at a single node, including behaviors, physiology, emotions, fatigue, environments, and locations, so as to enable Soldier-BSNs to obtain sufficient evidence, decision-making ability, and information resilience under resource constraints. In addition, we systematically discuss the problems and solutions of each unit according to the frame structure to identify research directions for the development of wearable devices for the military.

## 1. Introduction

The interconnection and collaborative decision-making between combat equipment and battlefield resources is one of the characteristics of the Internet of Battlefield Things (IoBT) [1]. The concept of the IoBT was proposed by the U.S. Army Research Laboratory (ARL) to enable predictive analytics for intelligent command and control and battlefield services. The next generation of military networks will consist of densely deployed battlefield nodes, including weapons, vehicles, robots, and human wearable devices. Command, control, communications, and intelligence (C3I) systems will accelerate integration with the IoBT to influence military decision-making in future wars [2]. As an important component of the IoBT, soldiers are the most flexible information nodes on the battlefield. Heterogeneous sensors integrated into a soldier’s equipment provide a command center with multidimensional battlefield information. Meanwhile, as an interdependent and interconnected group of entities, soldiers constantly communicate, coordinate, and jointly plan and execute tasks using the equipment of the IoBT.

Wearable sensors are the basic elements of military smart devices. The sensing module, which has information resilience, collects information on the battlefield under strict resource constraints, and realizes data transmission and analysis by cooperating with other modules that are integrated in the equipment. Despite its focus on improving situational awareness, the military wearable sensor network is an extension of Body Sensor Networks (BSNs). As one of the components of the modern IoT, BSNs can realize information exchange between human beings and objects so as to monitor, analyze, and control the body and the surrounding environment [3]. As a special domain of BSNs, Soldier-BSNs (S-BSNs) have more stringent environmental constraints and greater security pressure. Researchers have attempted to encrypt the sensed data by various methods, such as relying on individual differences in Electrocardiogram (ECG) signals, to protect relevant information on soldiers [4]. With the development of S-BSNs, the IoBT will have more robust and more reliable information nodes. Meanwhile, it is important to design high-performance data fusion methods. In other words, information fusion is the best way to broaden the functional boundary of BSNs and to improve the resilience of the network.

A robust military sensing system should process raw data in a multi-source fusion manner, including classification, tracking, decision-making, prediction, and optimization. In recent years, researchers have proposed specific solutions for sensor network optimization [5], decision fusion optimization [6], data redundancy processing [7], workflow scheduling [8], and network resilience in the distributed mode. In particular, the authors in [9] propose an integrated intelligent system based on swarm logic that integrates high-level data analysis with low-level device/task management. In addition, with improvements in computing and storage capacity, it is inevitable that an autonomous or semi-autonomous analysis platform (Fusion Center) will be established based on artificial intelligence (AI) technology. In the field of visual tracking, machine learning has become an important tool for video surveillance, understanding scenarios, and human–computer interaction [10,11]. The authors in [12] propose a critical AI technology analysis to derive guidelines for designing distributed systems of Collective Situational Understanding (CSU). Their discussion on the processing of uncertain and sparse data is also helpful to the design of S-BSNs.

In this paper, we build a new fusion framework from an application perspective. The framework integrates low-level sensing data and high-level decision information to build a comprehensive and extensible application research system for S-BSNs. Our contributions are twofold. Firstly, a multi-level fusion framework (MLFF) is proposed. Compared with a battlefield situation, we are more concerned with the physiology of a soldier, which is a neglected part of current military smart devices. Previous research on wearable technology mainly focuses on disease diagnosis [13,14], physical rehabilitation [15], and action recognition [16] in non-military fields. However, these techniques are not suitable for highly trained soldiers on the battlefield. Considering the complexity of the battlefield environment, we define the attributes of soldiers in terms of three levels. We encourage high-level physiological information to be analyzed in the multi-source fusion mode to fully utilize sensor resources. The MLFF involves some recessive physiological states, such as emotion and fatigue, which will be combined with other necessary physiological parameters to construct a new evidence set to analyze more complex physiological information. Secondly, according to the structure of the proposed framework, we summarize the future research directions and specific problems to be solved for each system unit in a battlefield environment, and provide a corresponding reference scheme based on the latest research results. We believe that the development of wearable devices for the military should be based on the safety of soldiers. Only by reducing the number of casualties that are caused by injuries or other emergencies on the battlefield can the success rate of military missions be improved. Therefore, research directions on the physiological monitoring of, emotion recognition in, and behavior tracking for soldiers have been proposed. We also encourage the design of integrated systems for soldier safety that combine environmental information and location technology.

The rest of the paper is organized as follows. Section 2 introduces the latest developments in the integration of military smart equipment and multiple sensors and summarizes review articles on wearable technology in other fields. Section 3 describes the proposed framework and the model of the deployment of heterogeneous sensor networks. Section 4 summarizes some research directions and specific problems to be solved for each system unit in the battlefield environment, and provides a corresponding reference scheme based on the latest research results. Section 5 contains further summaries and discussions.

## 2. Related Work

The primary goal of the IoBT is to integrate new sensing platforms into military activities. Before the concept of the IoBT was proposed, researchers had started to design modern soldier equipment that was equipped with multiple sensors [17]. Although most of the research was aimed at increasing resource competitiveness and situational control in battlefield confrontations, the security of soldiers has also been increased. The U.S. Land Warrior Integrated Soldier System (LWISS), which includes a protective system, an information-processing system, and a combat system, has been one of the outstanding achievements of military smart devices [18]. The system integrates a variety of sensors and communication devices that enable soldiers to exchange intelligence in the point-to-point mode. The U.S. Tactical Assault Light Operator Suit (TALOS), which has been in development for many years, is equipped with exoskeletons, smart helmets, protective armor, and multiple biosensors to maximize the battlefield capabilities of soldiers. Some of the latest research results on wearable devices for the military are shown in Table 1. It can be seen that the next generation of smart wearable devices for the military will address soldiers’ task efficiency. Research on assistive systems based on physiological parameters is the key to upgrading arms.

With the development of wearable sensors towards miniaturization and intelligence, physiological analysis based on multiple heterogeneous sources has been applied in many fields. We have summarized the latest review articles on research achievements in wearable technology. Although these reviews do not cover the military field, the research findings that are discussed are useful for developing and upgrading S-BSNs. Firstly, according to a functional classification, Ultra-Wide Band (UWB) technology [19], motion monitoring [20], medical rehabilitation [21,22,23], gait analysis [24], and fall detection [25,26] have been comprehensively discussed by previous scholars. UWB has always been one of the key technologies for battlefield positioning. A modularized UWB device can realize short-distance positioning for soldiers in a global positioning system (GPS)-denied environment. Similarly, research results on behavior recognition and physiological monitoring under battlefield constraints can be applied to the development goal for S-BSNs. Secondly, according to a classification by the wearable device’s position on the body, some reviews have discussed wearable devices based on the wrist [27], the hand [28], textiles (the body) [29,30], and the exoskeleton (upper limb [31] and lower limb [32]). Military exoskeletons and intelligent textiles have always been the focus of military equipment researchers. The monitoring of physiological signals, the improvement of basic capabilities (load or stadia), and the provision of interfaces for human–computer interaction are realized by densely deployed heterogeneous sensors. Therefore, this kind of review may be valuable for researchers who are concerned with local physiological parameters. Thirdly, considering that soldiers comprise a population of the right age and with strong bodies, we believe that researchers should explore the age differences in target groups to distinguish the physiological model parameters of soldiers from those of the elderly [33,34] and infants [35,36]. Finally, although this paper does not discuss methods for the fusion of sensors and parallel system units, we have selected two review articles to help scholars who are interested in wearable technology based on multi-sensor fusion [37,38].

In this paper, a multi-level fusion framework (MLFF) based on S-BSNs is proposed. Considering the complexity of battlefield constraints, we discuss research directions for and the specific problems of each system unit in detail, and summarize the corresponding reference schemes for each problem. The proposed framework focuses more on the personal attributes of soldiers, while retaining the traditional sensing capabilities, to emphasize the importance of soldiers’ safety. To the best of our knowledge, an insufficient amount of research has been done on wearable technology based on information fusion in the military field. Therefore, this paper aims to establish a generalized research system for S-BSNs, and provide references to researchers by analyzing specific problems. It should be pointed out that, since information fusion has gradually developed into a hybrid technology that involves self-learning, self-organization, and self-adaptation, and since high-performance fusion algorithms and new theories on fusion continue to emerge, we will systematically analyze methods for information fusion, data coordination mechanisms, and algorithm nesting patterns based on the multi-level fusion framework in future work to further improve the decision-making performance of wearable devices for the military.

## 3. Multi-Level Fusion Framework (MLFF)

There is an unknown correlation among the behavior, physiology, and psychology of soldiers. In complex battlefield environments, sensing modules have a very limited role in tracking single vital signs to monitor the status of soldiers is very. Multi-sensor information fusion is a commonly used method to improve the accuracy and robustness of systems. In this section, a multi-level fusion framework for S-BSNs is proposed that aims to obtain more types of information on the safety of soldiers with higher reliability under resource constraints.

### 3.1. The Composition of MLFF

Firstly, extreme emergencies (C-level information) that involve soldiers in a battlefield environment, such as injuries, asphyxiation, and disappearances, need to be reported in a timely manner. Therefore, the ability of the system to identify and to respond to emergencies is particularly important. However, complex events are often accompanied by strong fluctuations in physiological signals. The fusion of environmental information, location information, and the results of an analysis of multiple physiological states can effectively improve the accuracy of an emergency’s identification and can reduce the number of false alarm events that sensor faults cause. Secondly, independent analysis systems (B-level information), including a Fatigue Detection System (FDS), an Emotion Recognition System (ERS), a Behavior-Tracking System (BTS), an Environmental Detection System (EDS), a Cooperative Localization System (CLS), and a B-level physiological parameter set are necessary for calculation, storage, and invocation. The results of an analysis from an independent system can not only serve as cognitive evidence of C-level information, but also provide feedback to the discrete state of soldiers. B-level information provides sufficient information resilience for S-BSNs. Since the results of information fusion in the system are interrelated, the final decision will not depend only on the sensor with the worst performance. Finally, as the basic elements of the MLFF, homogeneous or heterogeneous sensors provide A-level information for the system. The number of sensors and their type and location are determined by the integrated system functions of the equipment. Figure 1 shows the multi-sensor based MLFF model, which integrates six B-level information systems and involves 16 kinds of heterogeneous sensors.

Structurally, the MLFF has three layers. For B-level information, multiple pieces of A-level evidence are required to support the results of fusion analysis in each system. In other words, the output of each sensor may affect the results of multiple systems. This pattern of cross-fusion ensures that any B-level system will not feedback abnormal events separately. Considering the complexity of C-level events, decision-level fusion is more suitable for the process of analyzing an emergency. The structural advantage of the MLFF enables the system to increase the amount of evidence in decision-level fusion under the condition of limited sensor resources (including all B-level information and some A-level information related to weight, which will be explained in Section 4.2). For nodes of the same level, the robustness of A-level information depends on the number and deployment location of homogeneous sensors; the output results of B-level units are highly correlated, and the results of an analysis with a large deviation will conflict with a system of the same level. From the top view of the model, it can be seen that the MLFF collects basic evidence through a large number of sensors (homogeneous sensors are not shown in the figure) and gradually upgrades the results of analysis in the mode of cross-fusion, which will be used to identify emergencies involving soldiers’ safety. The MLFF always emphasizes the importance of soldiers’ safety. Therefore, no matter what method of information fusion researchers adopt, the accuracy of the recognition of a soldier’s status is always a priority.

From the perspective of the fusion center, the MLFF provides an integrated framework for existing wearable-technology-based multi-source fusion methods. Considering the computing power of the terminal and the pressure from the system’s communication bandwidth, data-level fusion is the most suitable way to solve conflicts in the data of homogeneous sources. For fusion with heterogeneous sensors, nonlinear mathematical methods with fault tolerance, adaptability, associative memory, and a parallel processing ability all meet the basic fusion requirements. In the MLFF, original data are calculated and stored according to the service objects in different data analysis modes (a terminal calculation or an upload to the cloud). In this process, an assessment of the results and sensor/task management are necessary. High-level information is used to make a judgment on the accuracy of a system of the same level and feedback the human body’s state according to the reasoning mechanism. In addition, a reliable fusion system needs to ensure the timely registration of information on interactions and the synchronization of the interface of the sensing–analysis–feedback–optimization process.

### 3.2. Sensor Deployment Model Based on the MLFF

Taking the system type shown in Figure 1 as an example, the S-BSNs will be composed of six B-level systems and multiple sensor nodes. Due to the existence of homogeneous sensors that serve different systems or the same sensing module, the number of sensors will continue to increase. We summarize the types of sensors that are used in various systems, and describe the model of the deployment of the wearable sensor network, in Figure 2 according to selected statistical results from the literature (the results from the latest 88 research articles are also used in the discussion of research directions in Section 4). It can be seen that A-level information is captured by the sensing nodes that cover key parts of the body, and homogeneous sensors that are deployed on different body parts may be responsible for data collection for different systems. It is worth mentioning that there may also be multiple homogeneous sensors for the same node, which can solve conflicts in data and improve the sensing module’s reliability by means of weighted fusion. Therefore, the capability of the smart device to integrate sensors will determine the final effect of the MLFF. In addition, according to the latest research results, some wearable devices for the military, such as smart glasses and military exoskeletons, have been added to Figure 2. These devices provide a platform for integrating more sensors and the possibility for S-BSNs to be equipped with more complex decision-making systems.

## 4. Systematic Analysis of Framework Units

In this section, the research directions for, and the specific problems of, each framework unit based on battlefield constraints will be systematically analyzed. For each problem, relevant research results from other fields are cited to provide references for the development of wearable technology for the military.

### 4.1. A-Level Information

A-level information refers to the original data from the sensing layer. Fusion of the data from homogeneous sensors is an effective method to improve the reliability of the sensing module. For fusion methods, multi-source synchronization and weight distribution are the basis of updates to algorithms. In addition, a conflict in data is a common problem in the fusion of multiple homogeneous sources [39]. Highly conflicting system inputs may lead to counterintuitive results. Sensor signals may couple different types of noise in complex battlefield environments; thus, conflicts in data are inevitable even after the same data preprocessing process has been performed. Improving the ability of the sensing module to integrate sensors and solving conflicts in data are important tasks at the A-level.

### 4.2. B-Level Information

B-level information is composed of information from multiple parallel systems. According to all of the possible states of a soldier in a battlefield environment, the targets of the detection system are divided into fatigue, emotion, behavior, location, surrounding environment, and some B-level physiological parameters, including blood pressure, oxygen saturation, respiration rate, and energy consumption. It should be noted that the B-level physiological parameters are calculated by the fusion of data from multiple biosensors (this is the difference between A-level data and B-level data). Very mature methods for data fusion are available, but we do not describe them in this paper.

#### 4.2.1. Fatigue Detection System (FDS)

In a military operation, the most common physical condition of a healthy soldier is fatigue. A soldier’s task efficiency decreases with physical fatigue. At present, this recessive physiological state cannot be fed back to the command center in time. The actual battlefield situation may deviate away from the expected situation as it develops due to a lack of coordination between soldiers and the command system. Therefore, it is necessary to integrate an FDS into smart devices for the military. The latest research focuses on the identification of driving under conditions of fatigue. The input to the system requires data from a combination of biosensors and the vehicle’s built-in equipment, such as the seat, steering wheel, and rearview mirror. For soldier wearables, portability is a key consideration. Therefore, we ignored research results that were obtained with complex sensing devices when searching for the reference schemes. Similarly, we did not consider research results that were obtained with a subjective questionnaire and subjective behavior calibration.

Signs of fatigue are concentrated in the muscles, skin, brain activity, and facial activity. The currently available detection technologies, irrespective of whether they analyze a single feature or fuse data from multiple sources, have limitations in a battlefield environment. The specific problems and corresponding reference schemes are shown in Table 2.

Muscle fatigue is a typical phenomenon in human body fatigue. First, soldiers on a battlefield tend to be in patterns of complex action, which result in multi-component and nonstationary Electromyogram (EMG) signals. Second, soldiers may ignore lower-level muscle fatigue, which can affect the precision of performed operations. Similarly, a mechanical analysis of muscles and an analysis of the state of deep muscles are key technologies in military FDSs. For the states of deep muscle, previous studies used invasive detection or EMG array acquisition and processing techniques [50]; however, such techniques are not applicable to military scenarios. Therefore, wearable technology for the non-invasive analysis of deep muscle needs further exploration. In terms of hardware, traditional EMG sensors often use multiple disposable electrodes, which not only increase the weight of wearable devices, but also fail to satisfy the application conditions. Finally, we consider that a complete FDS should realize some extended functions through external devices.

Smart helmets provide a platform for measuring Electroencephalogram (EEG) signals in real time. Previous research has shown that EEG signals contain multiple characteristic parameters that are related to physiological fatigue [51]. Similarly to the specific problems that EMG sensors have, multi-channel EEG sensors will reduce the robustness of the system. Moreover, since the human brain’s structure is directly related to the signal characteristics [52], it is important to select an appropriate position in the helmet to detect fatigue in soldiers. With the development of Augmented Reality technology, smart glasses integrated into helmets will provide soldiers with more information on the battlefield. However, researchers have found that fatigue from stereo vision can seriously affect cognitive ability. Military wearables researchers should take this type of fatigue seriously.

In addition, different types of physiological signals can be used to recognize fatigue. Although most facial activities are subjectively controllable, these facial features can still serve as a source of auxiliary information for the detection of fatigue under certain conditions and ensure the recognition of fatigue with high accuracy [53]. For example, evidence of fatigue can be obtained by tracking eyeball states [54,55]. Similarly, physiological signals, such as ECG signals [56], respiratory (RSP) signals [57], galvanic skin response (GSR) signals, and blood oxygen [58,59], provide possibilities for the design of an FDS based on information fusion. Information on the fatigue of soldiers is a form of recessive information. We need to continuously increase the amount of evidence to support fatigue detection in soldiers.

#### 4.2.2. Emotion Recognition System (ERS)

In general, soldiers’ emotions influence their decision-making on the battlefield. In an actual battlefield situation, negative emotions, such as anger, fear, and anxiety, can easily be aroused. In the classical multidimensional emotion space model, these negative emotions have similar features (all inside the zone with a negative valence and a high degree of arousal) [60]. So, an ERS must rely on physiological signals to distinguish negative emotions. Shu et al. [61] summarize the relationship between the features of various physiological signals and different emotions. We introduce negative emotion as a source of auxiliary information for the design of an ERS for the military, as shown in Figure 3.

We focus on several issues with an ERS for the military. Previous research has shown that movement can affect human emotions, meaning that an ERS needs to consider a larger number of complex factors, such as the motion of soldiers or individual physiological differences. The authors in [62] analyze the differences in emotional expression among people in a cross-cultural population from the perspective of visual representation. The results show that an ERS based on facial features cannot universally apply to a military with a multi-racial culture. The amount of information that is input into an emotion recognition system will inevitably need to be increased. As a special occupation with strict training, soldiers are more restrained in the emotional expression of physiological stimulation. If facial features are taken as the source of information, an ERS needs to have the ability to recognize micro-expressions and a higher sensitivity, namely the ability to recognize emotion in real time. Although emotional information can help the command center to determine the state of soldiers in a battlefield environment, negative emotions are often non-discrete. Precise emotional outcomes, such as compound emotions [63], require further investigation.

An ERS can call on a variety of data from sensors. Due to its being highly sensitive and a mature signal acquisition technology, and the low degree of subjective control, EEG signals are one of the most important types of physiological signal for the quantitative analysis of soldiers’ emotions. However, EEG signals have feature parameters with a high number of dimensions, and redundant features will reduce the efficiency of the algorithm. The design of the method for feature extraction and the selection of feature types are the key techniques for EEG signal analysis. In the data analysis process, considering the storage, calculation, and data transmission capacity of the equipment, we suggest reducing the data training time or the number of iterations when updating the algorithm on the premise of ensuring accurate recognition of negative emotion [64]. Different from physiological signals, such as EEG or ECG signals, facial representation plays an auxiliary role in the emotion recognition process. Some changes, such as the movement of the mouth or nose [65], have a certain reference value. Previous studies have shown that an ERS based on the fusion of data from multiple sources has higher accuracy. Even with a fast decision-level fusion method, the result is more reliable than that obtained with a single sensor [66,67]. The physiological parameters that were used in the latest research include EEG, ECG [68,69], GSR [70,71], EMG [72], PhotoPlethysmoGraphy (PPG) [73], body temperature [74], and facial features. The authors in [75] quantify human stress by combining heart rate variability (HRV) with salivary cortisol; however, the process for the extraction of biochemical signals is not suitable for passive monitoring of emotions. Similarly, the authors in [74,76] added Electrooculography (EOG) to the data fusion process. To our knowledge, no wearable device for non-invasive EOG measurement exists that can provide real-time physiological information, which poses a new challenge for researchers of wearable devices. Table 3 shows some of the specific problems of ERSs and the corresponding reference schemes.

#### 4.2.3. Behavior Tracking System (BTS)

In battlefield environments, soldiers may be in an unusual state of behavior, such as a rapid march, a high-altitude jump, and injury. A BTS can operate in parallel with other detection systems and provide more accurate body status information for medical units or the command center. Smart devices for the military, regardless of whether it is a robot on the battlefield as an independent sensing node or an intelligent exoskeleton that used to cross the boundaries of a soldier’s ability, need soldiers to provide multi-dimensional input to the system. The currently available behavioral recognition technology can use limb movements with significant characteristics to excite the control system, which means that BTSs play an important role in S-BSNs.

Considering the computational resources, we classify the behaviors of soldiers according to their complexity [86]. A simple movement refers to a basic behavior with obvious characteristics, such as standing, sitting, falling, walking, and running. The results on the recognition of behaviour are used to supplement the high-level information. For example, a body tilting behavior is divided into a rapid tilt (passive) and a slow tilt (active) [87]. When a soldier is in a rapid tilt, other physiological signals, such as blood pressure, heart rate, respiration, and pulse, can be used to remotely determine whether the soldier has fallen [88]. Similarly, multiple systems use gait characteristics in their analysis [89]; examples are parallel systems that recognize emotions, quantify energy loss, or identify soldiers’ identities [90]. Complex forms of movement focus on localized body behavior. A BTS covers a range of positions for wearable devices according to behavioral characteristics, such as a head controller [91,92], smart gloves [93,94], an exoskeleton for the waist [95,96,97], and smart shoes [98,99]. In addition, smart textiles provide possibilities for behavior recognition. The status of some key body parts, including the arms [100] and knees [101,102], needs to be tracked in real-time by inertial units. Table 4 summarizes research directions based on different body parts. It is worth mentioning that the analysis of simple behaviours and the analysis of complex behaviors have several issues in common that need to be considered. The first is energy loss [103]. A BTS is the main system that is used to calculate energy loss in soldiers. It can operate together with the FDS to determine a soldier’s working status, which helps the command center deploy soldiers. The second is the recognition of fast motion [104]. To the best of our knowledge, it is difficult for wearable sensors to recognize fast motion; however, it is a research issue that will inevitably arise in complex battlefield environments. The final issue is computing cost [105]. Compared with the other systems in the MLFF, the BTS has the largest number of sensors, which means that we need to conduct data mining on behavioral data with a high number of dimensions. The optimization of computing resources is an important problem in a BTS.

Different from the action recognition system for fixed scenes, we cannot arrange image-sensing devices in a battlefield environment based on a third-person perspective. Although image-processing technology has outstanding advantages in the field of action recognition [106,107,108], we have to rely on the classic inertial components as the main information acquisition device. Previous studies have shown that inertial sensors are highly sensitive to the speed and the angle of a limb’s movement, and these sensors have been used in motion pattern analysis [109] and gesture recognition [110,111]. EMG [112], ECG [111], and EEG [113] signals can also be used for behavior recognition. EMG signals are often used to obtain data on precision operations because of their ability to provide feedback on local limb movements. For example, the control methods that the authors in [114] discuss, which are based on electromyogram-pattern recognition (EMG-PR), provide possibilities to design the control interface of the exoskeleton. In addition, the authors in [115] designed a low-cost optical fiber force myography sensor to provide a new channel for non-invasive hand posture identification. Information fusion is the key technology of these systems. A BTS contains many homogeneous and heterogeneous sensors. The advantages of this structure are the the robustness and the accuracy it provides the system with. Complex noise types and the highly conflicting original data force us to adopt in a BTS for the military feature-level fusion (a process to reduce the dimensionality of the feature parameters needs to be considered) and decision-level fusion. As an important part of B-level information, the results from a BTS can be used in almost any multi-system fusion framework to determine soldiers’ more complex physiological states.

#### 4.2.4. Cooperative Localization System (CLS)

The development of communication and positioning technology is mutually reinforcing. The location of units is usually determined by a wide-area positioning system in force-based military operations, which helps to analyze the battlefield situation and with the exchange of intelligence. However, from the perspective of wearable devices, CLSs tend to feedback and modify the information from battlefields. In other words, these systems facilitate interaction between team members, such as maintaining a formation during special operations or providing the location of nearby wounded soldiers. This mechanism enables soldiers to construct a moving coordinate system in an interactive mode to mark a target’s position, thus reducing the response time in the case of an emergency. Therefore, wearable devices for the military should not only adopt wide-area positioning technology (such as a Global Navigation Satellite System (GNSS)), but also integrate a short-range CLS, which determines the type of sensor, the number of sensors, and the utilization efficiency of the sensors in S-BSNs.

Complex battlefield situations may cause soldiers to be located in an environment in which wide-area positioning technology does not work. The CLS needs to obtain the relative position of soldiers when GPS, base station, pseudolite [119], and other positioning technologies fail. The system’s design needs to consider the hardware structure, the positioning method, the positioning performance, and environmental interference, which are problems that indoor positioning technology also faces. Indoor positioning refers to short-range positioning technology in a fixed area, including infrared positioning [120], ultrasonic positioning [121], Radio Frequency Identification (RFID) positioning [122], UWB positioning [123], inertial navigation [124], geomagnetic positioning [125], and Visible Light communication (VLC) positioning [126]. The changing environmental constraints make it impossible for the CLS to adopt infrared positioning that relies on line-of-sight propagation and geomagnetic or WLAN (Wireless Local Area Network) positioning technology based on the fingerprint method. Moreover, the VLC technology cannot be used due to the sensitivity of the signal. Table 5 lists short-range positioning techniques that satisfy local battlefield positioning conditions. Considering the utilization efficiency of the sensor, the IMU (Inertial Measurement Unit) can be used as the core component without a global position tag. An inertial system that integrates auxiliary sensors, such as a barometer, can calculate the multi-dimensional coordinates of soldiers. However, with the iterative accumulation of calculation errors, the reliability of the system will be seriously reduced. The fusion of multiple positioning techniques is necessary to help correct inertial data in real time. UWB, ultrasound, and RFID positioning all require the integration of transceivers, which increase the load on the soldier. Although many studies have demonstrated that the UWB technology has significant advantages for battlefield localization, developing a micro-positioning module remains a challenge [19]. It is worth mentioning that the integration technology for the ultrasonic sensor is mature. Although ultrasound is susceptible to interference from the transmission medium, it has advantages in terms of penetration and positioning range when compared to RFID. The RFID positioning technology is greatly restrained in a battlefield environment. Passive RF can realize close positioning; however, it has no communication ability. Active RF has a relatively wide positioning range; however, it requires a sufficient energy supply. In a word, CLSs need to realize technology fusion under the condition of limited resources. Adding different types of basic sensing devices can improve the robustness of the positioning system.

#### 4.2.5. Environment Detection System (EDS)

The conventional battlefield environment detectors are a series of target-driven independent devices. The detection results are used to determine the degree of the threat that an antagonistic environment represents to soldiers in real time. Similarly, an EDS is composed of several wearable environmental sensors that are used to monitor the air temperature, air pressure, light intensity, radiation, and other variables in the environment around soldiers. Studies have shown that there is a correlation between environmental information and a human’s physiological state [128]. Moreover, miniaturized environmental sensors add various data types to S-BSNs. On this basis, environmental information can carry weight in the MLFF, and be directly used in the analysis of a soldier’s status. However, heterogeneous data types present challenges for the design of high-performance fusion methods. As far as we know, the fusion approach that involves environmental information is limited to decision-level fusion. However, this approach can still be used to upgrade the information in the framework. From the perspective of information fusion, the environment is not only an attribute that is used to model the battlefield, but also one of the valid forms of evidence that improves the information resilience of S-BSNs. In other words, the proposed framework has sufficient evidence to ensure real-time monitoring of a soldier’s physiological state.

### 4.3. C-Level Information

Different from commercial devices, wearable devices for the military need to consider emergencies involving soldiers’ safety in addition to analyzing basic physiological states. An emergency is composed of a series of basic events. We have systematically reviewed the latest research on basic events in different fields. The MLFF provides the possibility to integrate different analysis techniques and recognize more complex events.

An emergency is correlated with multiple physiological signals. For example, being shot can lead to blood loss and emotional fluctuations, and severe injuries may cause drastic changes in physiological parameters, such as blood pressure, blood oxygen, heart rate, and PPG [17]. Therefore, further fusion of B-level information is necessary. However, even with the most advanced computing and communication technologies that improve the real-time analysis capability of wearable devices, data mining in multi-dimensional heterogeneous information remains a challenge. Decision-level fusion is considered to be more suitable for S-BSNs with a deep fusion of data and technology. Compared with data-level and feature-level fusion, decision-level fusion can not only analyze independent system units in the abstract, but also reduce the calculation cost and the pressure on the communication bandwidth. Moreover, the MLFF provides more types of evidence for decision-level fusion, so as to reduce the impact of the low accuracy. By summarizing selected research results on wearable devices (88 papers), we present the relationship between all of the involved system units and research problems in the proposed framework, as shown in Figure 4. It can be seen that C-level information can be co-analyzed by multi-sensor, multi-method, and multi-level fusion. Meanwhile, the framework’s structure clearly shows the weight of each sensing module in the S-BSNs. In our statistics, ECG signals have a greater number of edges compared with other nodes, indicating that the analysis of various systems is more dependent on ECG signals. In other words, the more highly weighted a sensor’s output is, the more physiological information it contains. Therefore, we suggest that highly weighted A-level information be added to the evidence set of a C-level emergency to further increase the number of types of evidence for decision-level fusion and improve the resilience of the system on the battlefield.

We are concerned about the resilience of the S-BSNs on the battlefield. Since the MLFF is to be applied in the military domain, all systems must tolerate disruptions in communication, a loss of sources, and malicious data inputs [129]. Wearable devices for the military not only need a parallel data coordination mechanism, but must also ensure system robustness in the case of a loss of evidence. Within the range of a wearable device’s energy and computing capabilities, broadening the bounds of the feasibility of accurate reconstruction of the state of a system by increasing the number of types of evidence is the most effective way to establish a resilient fusion framework.

## 5. Discussion

As a key technology in battlefield situational awareness, information fusion has always been the focus of military research. However, we have to admit that there is still a big gap between the automatic fusion processing capability and the requirements of high-level applications. The traditional Joint Directors of Laboratories (JDL) model has developed into a multi-stage fusion model that is driven by a target/application that allows for cognitive assistance. The typical representative of this “human-in-the-loop” pattern is the Data Fusion Information Group (DFIG) model. However, the service objective of the JDL/DFIG model in the military field always focuses on wide-area information, such as situational awareness or target tracking, and improves the decision-making ability of the system by emphasizing the interaction between users’ active behaviors and each stage of the model. In contrast, the MLFF is designed for S-BSNs. It is regarded as a fusion model for inward-facing sensing of a single node in the IoBT. In other words, users are more involved in the fusion process at all levels in a passive manner. At the same time, high-level decisions can still be optimized using a “user (or command center) refinement” approach. In addition, unlike the JDL model, which must occur in a hierarchical order, the structural characteristics of the MLFF provide the possibility for low-level framework units (source data management) to directly participate in body state assessment. Table 6 shows a comparison between the MLFF and some of the existing frameworks/systems. It can be seen that the proposed framework also has certain advantages in terms of function coverage.

In Section 4, we systematically analyzed the research questions at each level according to the proposed framework. However, wearable technology for the military based on information fusion also faces many challenges in framework implementation. First of all, the complex sensor deployment puts tremendous pressure on the device’s energy and body load. The MLFF supports the integration of more types of sensing modules; however, the upper limit of energy storage is positively correlated with the weight of wearable devices. Although some teams have started to pay attention to the design of power modules, such as SPaRK (energy-scavenging exoskeletons) described in Table 1, improving the energy storage capacity and reducing soldiers’ loads remain a challenge. Secondly, in terms of communication and computing, multi-dimensional heterogeneous data need a reasonable communication bandwidth and computing mode. Collaboration between edge computing and cloud computing is the core technology of data analysis. In a complex battlefield environment, a multi-sensor, multi-method, and multi-level analysis framework not only requires a centralized processing mode to manage heterogeneous sources and provide a knowledge base or artificial bias for intelligent decision-making, but also needs a terminal computing capacity to improve the communication performance and decision-making efficiency. Finally, information security in antagonistic environments is more complex with the diversification of communication and data storage patterns. As one of the open challenges of the IoBT, information security has always been one premise for the development of wearable technology for the military.

## 6. Conclusions

With the development of the IoBT, the intelligent upgrade of S-BSNs has become inevitable. While retaining situational awareness of the battlefield, the next generation of wearable devices for the military should be based on the personal attributes of soldiers. Although highly integrated systems can meet the diverse service needs of modern battlefields, a complete fusion framework is necessary to adjust the cooperative relationship among all sensing modules and applications. Therefore, in this paper, a multi-level fusion framework based on heterogeneous sensor networks is proposed. The framework is divided into three levels, including original information collected by densely deployed sensing modules, cross-fused independent systems, and emergencies as determined from multi-dimensional evidence. The structure of the MLFF can increase the number of evidence types and the information resilience of S-BSNs under resource constraints. A final decision will not depend on the sensing unit with the worst performance. The proposed framework covers information on soldiers in multiple respects, including motion, physiology, emotion, fatigue, environment, and location, and allows for the addition of other types of sensors and systems. All decisions the system makes are used to identify and alert soldiers to extreme events involving their safety. In addition, since wearable technology based on multi-sensor fusion lacks a generalized research system in the military field, we sorted the specific problems of each system according to the proposed framework, and provided some research directions for the development of S-BSNs by referring to the research results on wearable technology in other fields.

We have built an application-oriented fusion framework for wearable devices for the military. However, the framework has some limitations. Firstly, there is an optimization problem in sensor networks. Although the functions of homogeneous and heterogeneous sensors are different, the occupied network resources cannot be ignored. Under strict resource constraints, the design of path-first and resource-first sensor deployment models needs to be further considered. Secondly, the system is not comprehensive enough. We have made an attempt to best meet the application needs of a healthy soldier in a battlefield environment, but the framework still needs to consider more functional modules. The existing applications also need to be improved. For example, in the ERS, in order to reduce the computational complexity, we only focused on the negative emotions of soldiers, which covers only half of the emotional space. Finally, the fusion framework needs to be implemented by nesting the existing multi-class application algorithms. In future work, we will systematically analyze the fusion methods that are used at each level, and propose a data coordination mechanism and an algorithm nesting pattern suitable for the MLFF to improve the decision-making performance of wearable devices for the military.

## Figures and Tables

**Figure 1 sensors-19-02651-f001:**
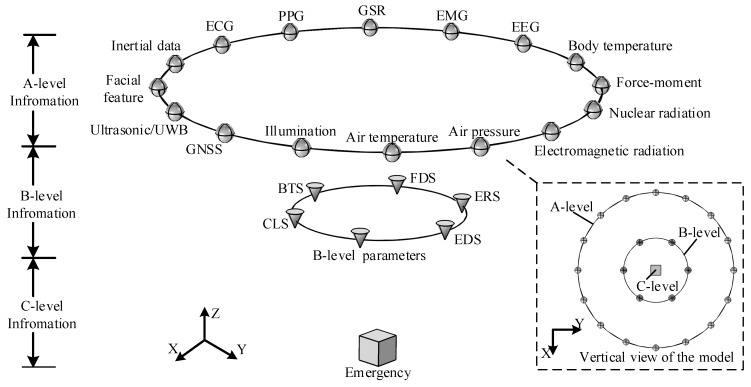
The multi-level fusion framework.

**Figure 2 sensors-19-02651-f002:**
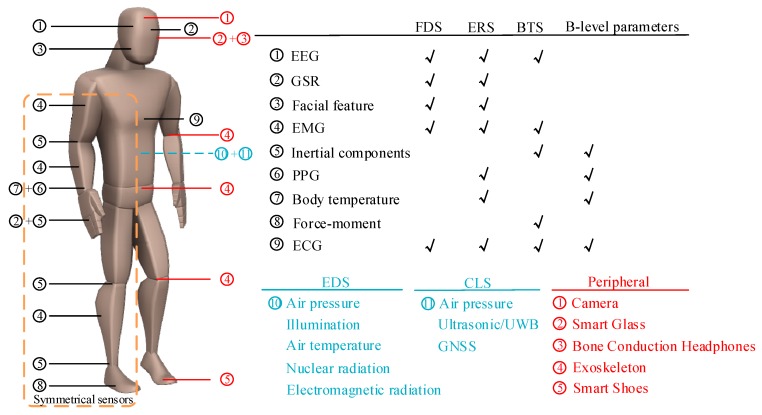
The sensor deployment model based on the multi-level fusion framework (MLFF).

**Figure 3 sensors-19-02651-f003:**
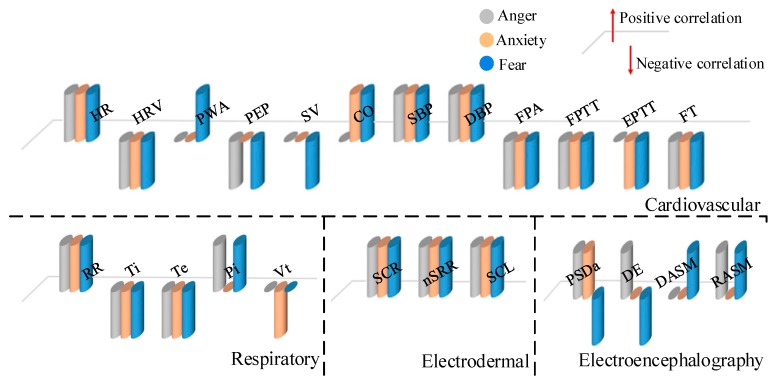
The correlation between physiological signal characteristics and three negative emotions.

**Figure 4 sensors-19-02651-f004:**
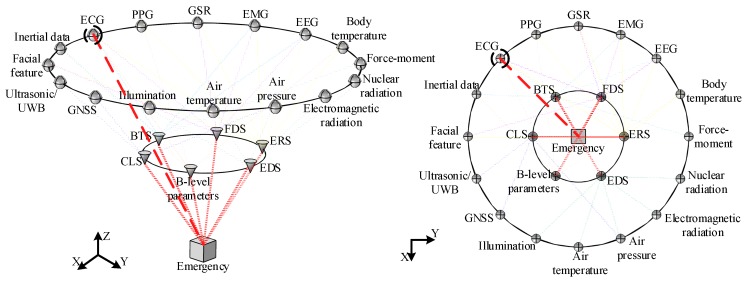
The relationships among all of the framework units are determined based on a summary of the latest research results on wearable technology.

**Table 1 sensors-19-02651-t001:** Existing research achievements on wearable devices for soldiers.

Name	Research Unit	Type	Characteristic
LifeBEAM	LifeBEAM	Helmet	Uses an optical sensor to measure heart rate
CombatConnect	The Army’s Program Executive Office	Wearable electronics system	Distributes data and power to and from devices via a smart hub integrated into the vest or plate carrier
Black Hornet 3	FLIR SYSTEM	Pocket-sized unmanned helicopter	An integrated camera that can be mounted on a squad member’s combat vest as an elevated set of binoculars
Ground Warfare Acoustical Combat System	Gwacs Defense	A wearable tactical system	Identifies and locates hostile fire; detects and tracks Small UAVs
ExoAtlet	ExoAtlet	Lower body powered exoskeletons	Provides mobility assistance and decreases the metabolic cost of movement
SPaRK	SpringActive	Energy-scavenging exoskeletons	The collected energy can be turned into electricity to recharge a battery or directly power a device

**Table 2 sensors-19-02651-t002:** Research directions for and the specific problems of Fatigue Detection Systems (FDSs).

Research Direction	Specific Problem	Reference Scheme	Physiological Signal
Signal processing	Multi-component and nonstationary signals	[40]	EMG
Information mining	Low-level muscle fatigue	[41]	EMG
	Local muscle analysis	[42]	EMG
	Stereoscopic visual fatigue	[43]	EEG
Sensor optimization	Multichannel detection	[44,45]	EMG; EEG
	Disposable electrode	[46]	EMG
	Detecting position	[47]	EEG
Function extension	Unloaded muscle effort	[48]	EMG
	Muscle recovery	[49]	EMG

**Table 3 sensors-19-02651-t003:** Research directions for and the specific problems of Emotion Recognition Systems (ERSs).

Research Direction	Specific Problem	Reference Scheme	Physiological Signal
Signal processing	Time frequency analysis	[77,78]	EEG
	Feature extraction	[60,79,80]	EEG
Information mining	Influence of movement	[72]	EMG, ECG, GSR
	Individual differences	[73]	ECG, GSR, PPG
	Cross-cultural differences	[62]	Facial features
	Micro-expression	[81,82]	Facial features
	Real-time recognition	[83]	ECG
Sensor optimization	Multichannel detection	[84]	EEG
Function extension	Control interface	[85]	ECG, EEG

**Table 4 sensors-19-02651-t004:** Research problems based on different body parts.

Body Parts	Function	Reference Scheme	Specific Problem	Sensor
Head	Monitoring	[116]	Movement characteristic	Inertial components
	Extension	[91,92]	Man–machine interaction	BNO 055 orientation modules; Six DOF position sensor
Hand	Monitoring	[94]	Hand function evaluation	Inertial components
	Extension	[93,115]	Man–machine interaction	Inertial components; Optical fiber force myography sensor
Arm	Monitoring	[100]	Movement characteristic	Inertial components
	Extension	[114,117]	Man–machine interaction	EMG sensor; Pressure sensor
Waist	Extension	[95,96,97]	Muscle fatigue and injury	Exoskeleton
Lower limb	Monitoring	[101]	Movement characteristic	Inertial components
	Monitoring	[102]	Knee load	Inertial components
Foot	Monitoring	[90]	Gait recognition	Inertial components
	Extension	[118]	Injury (GRFs)	3D force/moment sensors

**Table 5 sensors-19-02651-t005:** Short-range positioning techniques that satisfy local battlefield positioning conditions.

	Range	Error	Cost	Applicability	Restriction
Inertial components	1–100 m	<1% [124]	Low	Strong anti-interference ability; high utilization rate	Data processing; errors accumulate over time
Ultrasonic	1–10 m	<20.2 cm [127]	High	Correction of inertial data; improvement of relative coordinates	Signal attenuates significantly in harsh environments
UWB	1–50 m	<2 cm [123]	High	High penetration; high precision	Miniaturization of positioning devices
RFID	1–50 m	<10 cm [122]	Low	Strong anti-interference ability	Hard to integrate with other systems

**Table 6 sensors-19-02651-t006:** Comparison between the MLFF and existing systems.

	Physiology	ERS	FDS	BTS	CLS	EDS
MLFF	√	√	√	√	√	√
LWISS [18]	√	⨯	⨯	√	√	⨯
FlexiGuard [130]	√	√	√	√	⨯	√
CAPCoS [131]	√	√	⨯	⨯	⨯	√
AMS [132]	⨯	√	√	√	⨯	√

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
