# Peer review of "Systematic Analysis of a Military Wearable Device Based on a Multi-Level Fusion Framework: Research Directions"

_sensors, 2019, doi:10.3390/s19122651_

Round 1

Reviewer 1 Report

In this paper, the authors propose a multi-level fusion framework (MLFF) based on soldier Body Sensor Networks, and describes the deployment model of heterogeneous sensor networks based on the application system, it is interesting, but minor revision is still needed to match the requirements of the Journal:

1. the abstract can be rewritten to be more meaningful. The authors should add more details about their contributions in the field of military and BSN, the abstract should clarify what is exactly proposed and how the proposed approach is validated.

2. Quality of figures is so important too. Please provide some high-resolution figures. Some figures have a poor resolution

3. The novelty of this paper is not clear. What is the motivation of the proposed work? Research gaps, objectives of the proposed work should be clearly justified by citing the paper listed below:

1) Multi-Objective Workflow Scheduling With Deep-Q-Network-Based Multi-Agent Reinforcement Learning, IEEE Access, vol. 7, pp. 39974-39982, 2019.

2) Quantitative Assessment for Self-Tracking of Acute Stress based on Triangulation Principle in a Wearable Sensor System, IEEE Journal of Biomedical and Health Informatics 2018, 23(2):1-11.

3) Spatially regularized structural support vector machine for robust visual tracking. IEEE Transactions on Neural Networks and Learning Systems, 2018.

4) Heart-Beats Based Biometric Random Binary Sequences Generation to Secure Wireless Body Sensor Networks,  IEEE Transactions on Biomedical Engineering 2018, 65(12):2751-2759.

5) Multiple Kernel Coupled Projections for Domain Adaptive Dictionary Learning, IEEE Transactions on Multimedia, 2019.

6) Assessment of Biofeedback Training for Emotion Management through Wearable Textile Physiological Monitoring System, IEEE Sensors Journal 2015, 15(12): 7087-7094

4. the language usage throughout this paper need to be improved, the author should do some proofreading on it. Give the article a mild language revision to get rid of few complex sentences that hinder readability and eradicate typo erros.

5. What are the limitations of this work? How can the rigor of this work be demonstrated?

Author Response

Response to Reviewer 1 Comments

General Comments:

In this paper, the authors propose a multi-level fusion framework (MLFF) based on soldier Body Sensor Networks, and describes the deployment model of heterogeneous sensor networks based on the application system, it is interesting, but minor revision is still needed to match the requirements of the Journal:

Point 1: The abstract can be rewritten to be more meaningful. The authors should add more details about their contributions in the field of military and BSN, the abstract should clarify what is exactly proposed and how the proposed approach is validated.

Response 1: We appreciate your comment, and we have revised the abstract. In the revised version, we added more details about the article contribution and described in detail the service objects and meaning of the proposed framework. (In abstract, lines 19-24, page 1)

Point 2: Quality of figures is so important too. Please provide some high-resolution figures. Some figures have a poor resolution

Response 2: Thanks for noticing us this problem and we have revised all the pictures in this article to improve the clarity and readability of the pictures. (Figure 1-Figure 4).

Point 3: The novelty of this paper is not clear. What is the motivation of the proposed work? Research gaps, objectives of the proposed work should be clearly justified by citing the paper listed below:

1) Multi-Objective Workflow Scheduling With Deep-Q-Network-Based Multi-Agent Reinforcement Learning, IEEE Access, vol. 7, pp. 39974-39982, 2019.

2) Quantitative Assessment for Self-Tracking of Acute Stress based on Triangulation Principle in a Wearable Sensor System, IEEE Journal of Biomedical and Health Informatics 2018, 23(2):1-11.

3) Spatially regularized structural support vector machine for robust visual tracking. IEEE Transactions on Neural Networks and Learning Systems, 2018.

4) Heart-Beats Based Biometric Random Binary Sequences Generation to Secure Wireless Body Sensor Networks, IEEE Transactions on Biomedical Engineering 2018, 65(12):2751-2759.

5) Multiple Kernel Coupled Projections for Domain Adaptive Dictionary Learning, IEEE Transactions on Multimedia, 2019.

6) Assessment of Biofeedback Training for Emotion Management through Wearable Textile Physiological Monitoring System, IEEE Sensors Journal 2015, 15(12): 7087-7094

Response 3: We appreciate your comment. We have carefully studied the above articles and cited them in the paper. In recent years, wearable technology based on information fusion has developed rapidly. The integration of existing research results is of great significance to the development of military smart wearable devices. In this paper, we build a new fusion framework from the perspective of application. The framework integrates low-level sensing data and high-level decision information to build a comprehensive and extensible application research system for S-BSNs.

Point 4: The language usage throughout this paper need to be improved, the author should do some proofreading on it. Give the article a mild language revision to get rid of few complex sentences that hinder readability and eradicate typo erros.

Response 4: Thanks for noticing us these issues and we have revised the language throughout this article. We reduced complex sentences and some typos we found. Revisions have been marked by using the "Track Changes" function in Microsoft Word.

Point 5: What are the limitations of this work? How can the rigor of this work be demonstrated?

Response 5: We appreciate your comment and we reorganized section 5 into discussion and conclusion. In the conclusion, we analyzed the limitations of our work in detail (In conclusion, lines 532-545, page 16). In discussion, we compared our work with existing models or systems, and described specific challenges to the proposed framework. In section 3, we added a description of the framework implementation (In 3.1, lines 190-201, page 5). We hope to provide new research directions for the military field by analyzing wearable technologies in other fields, and we are working on the development of task/device management models and algorithmic nesting patterns based on the proposed framework.

Special thanks to you for your good comments.

Reviewer 2 Report

As a general comment, the present work represent a good high-level contribution to the definition of a sensor fusion framework with application to wearable Internet of Battlefield things. However, before publication, it is my opinion that the following major comments should be addressed by the authors:

1)      Abstract – “although the densely deployed wearable sensors” -> although the densely-deployed wearable sensors”

2)      Abstract – “still lack a generalized research system” -> “still lacks a generalized research system”

3)      Sec. I – “Meanwhile, soldiers,…” -> “Meanwhile, soldiers,

4)      Sec. I – “the Soldier-BSNs (S-BSNs) has more stringent environmental constraints and greater security pressure, but its functions are more complex.” -> the Soldier-BSNs (S-BSNs) have more stringent environmental constraints and greater security pressure, but their functions are more complex.”

5)      The following closely-related works on sensor fusion in surveillance/military applications have been missed by the authors. They could be used to provide a wider and richer introduction to the context of massive sensor use in military applications and the need for their effective capitalization, through information fusion, to get situation awareness:

[R1] Mobile sensor networks based on autonomous platforms for homeland security." 2012 Tyrrhenian Workshop on Advances in Radar and Remote Sensing (TyWRRS). IEEE, 2012.

[R2] "Learning and reasoning in complex coalition information environments: a critical analysis." 2018 21st International Conference on Information Fusion (FUSION). IEEE, 2018.

[R3] "Distributed classification of multiple moving targets with binary wireless sensor networks." 14th International Conference on Information Fusion. IEEE, 2011.

[R4] "Accurate and Timely Situation Awareness Retrieval from a Bandwidth Constrained Camera Network." 2017 IEEE 14th International Conference on Mobile Ad Hoc and Sensor Systems (MASS). IEEE, 2017.

[R5] "Quantizer design for generalized locally optimum detectors in wireless sensor networks." IEEE Wireless Communications Letters 7.2 (2018): 162-165.

6)      By reading the whole paper, it seems that the information fusion aspect (e.g. how the information arising from different heterogeneous body sensors are effectively integrated and capitalized) is only marginally/tangentially touched.

7)      I would like the authors to compare and discuss the considered MLFF in comparison to the well-known JDL/DFIG model.

8)      Please improve the rendering/readability of Fig. 4.

9)      Sec. 5 should be rewritten so as to also point to open challenges in the context of information fusion and wereable IoBT.

Author Response

Response to Reviewer 2 Comments

General Comments:

As a general comment, the present work represent a good high-level contribution to the definition of a sensor fusion framework with application to wearable Internet of Battlefield things. However, before publication, it is my opinion that the following major comments should be addressed by the authors:

Point 1: Abstract – “although the densely deployed wearable sensors” -> “although the densely-deployed wearable sensors”

Response 1: Thanks for noticing us this problem and we have rephrased this sentence in the abstract. “although the densely-deployed wearable sensors provide a platform for comprehensively monitoring the status of soldiers” (in abstract, lines 15, page 1).

Point 2: Abstract – “still lack ageneralized research system” -> “still lacks ageneralized research system”

Response 2: Thanks for noticing us this problem and we have rephrased this sentence in the abstract. “the wearable technology based on multi-source fusion still lacks a generalized research system to highlight the advantages of heterogeneous sensor networks and information fusion” (in abstract, lines 16, page 1).

Point 3: Sec. I – “Meanwhile, soldiers,…” -> “Meanwhile, soldiers,

Response 3: Thanks for noticing us this problem and we have reorganized the sentence.Meanwhile, as an interdependent and interconnected entity group, soldiers constantly communicate, coordinate, jointly plan and execute tasks by the equipment of the IoBT.”  (in introduction, lines 39 and 40, page 1).

Point 4: Sec. I – “the Soldier-BSNs (S-BSNs) has more stringent environmental constraints and greater security pressure, but its functions are more complex.” ->“the Soldier-BSNs (S-BSNs) have more stringent environmental constraints and greater security pressure, but their functions are more complex.”

Response 4: Thanks for noticing us this problem. We have deleted this sentence and made content modification. “As a special domain of BSNs, the Soldier-BSNs (S-BSNs) has more stringent environmental constraints and greater security pressure.” (in introduction, lines 48, page 2).

Point 5: The following closely-related works on sensor fusion in surveillance/military applications have been missed by the authors. They could be used to provide a wider and richer introduction to the context of massive sensor use in military applications and the need for their effective capitalization, through information fusion, to get situation awareness:

[R1] Mobile sensor networks based on autonomous platforms for homeland security." 2012 Tyrrhenian Workshop on Advances in Radar and Remote Sensing (TyWRRS). IEEE, 2012.

[R2]"Learning and reasoning in complex coalition information environments: a critical analysis." 2018 21st International Conference on Information Fusion (FUSION). IEEE, 2018.

[R3]"Distributed classification of multiple moving targets with binary wireless sensor networks." 14th International Conference on Information Fusion. IEEE, 2011.

[R4] "Accurate and Timely Situation Awareness Retrieval from a Bandwidth Constrained Camera Network." 2017 IEEE 14th International Conference on Mobile Ad Hoc and Sensor Systems (MASS). IEEE, 2017.

[R5]"Quantizer design for generalized locally optimum detectors in wireless sensor networks." IEEE Wireless Communications Letters 7.2 (2018): 162-165.

Response 5: We appreciate your comment. We have carefully studied the above papers and cited them in the introduction. These articles are helpful for us to understand multi-sensor-based situational awareness, scene understanding and target identification/tracking in the military field.

Point 6: By reading the whole paper, it seems that the information fusion aspect (e.g. how the information arising from different heterogeneous body sensors are effectively integrated and capitalized) is only marginally/tangentially touched.

Response 6: We appreciate your comment and we have added a description of the framework implementation in section 3 from the perspective of information fusion (In 3.1, lines 190-201, page 5). During the conception of this paper, we reduced the description of specific fusion algorithm or analysis method. Our work is to provide an application-oriented fusion framework for traditional multi-level fusion algorithms (data-level, feature-level and decision-level). In this paper, we systematically described the structure, working pattern, and service objects of the proposed framework. Next, we need to design a reasonable algorithm nesting pattern and task/device management system for the proposed framework, which will discuss the specific ways of information fusion and capitalization in detail.

Point 7: I would like the authors to compare and discuss the considered MLFF in comparison to the well-known JDL/DFIG model.

Response 7: In discussion, we have made a detailed comparison between JDL/DFIG model and the proposed framework (in discussion, lines 479-491, page 13).

Point 8: Please improve the rendering/readability of Fig. 4.

Response 8: Thanks for noticing us this problem and we have revised all the pictures in this paper to improve the clarity and readability of the pictures. (Figure 1-Figure 4).

Point 9: Sec. 5 should be rewritten so as to also point to open challenges in the context of information fusion and wereable IoBT.

Response 9: We appreciate your comment and we reorganized section 5 into discussion and conclusion. In the discussion, we have described the challenges of military wearable technology based on information fusion in terms of energy, weight, communication, computing, and information security (in discussion, lines 497-512, page 13 and 14).

Special thanks to you for your good comments.

Round 2

Reviewer 1 Report

The authors have revised the article according to my comments, it can be accepted to publish in current form.                

Reviewer 2 Report

As a general comment, the present work represent a good high-level contribution to the definition of a sensor fusion framework with application to wearable Internet of Battlefield things.

Additionally, the authors have satisfactorily addressed the comments I have raised in the previous review round and modified the manuscript accordingly. Therefore, I am glad to recommend the present paper for publication.